# Transcriptome Analysis Uncovers the Gene Expression Profile of *Hemileia vastatrix* (Race XXXIII) during the Interactions with Resistant and Susceptible Coffee

Isabel Samila Lima Castro [1,†], Rejane do Livramento Freitas-Lopes [1,†], Sávio de Siqueira Ferreira [2,†], Talles Eduardo Ferreira Maciel [1], Juan Carlos Florez [1,3], Eunize Maciel Zambolim [1], Laércio Zambolim [1] and Eveline Teixeira Caixeta [1,4,*]

1 Instituto de Biotecnologia Aplicada à Agropecuária (BIOAGRO), Universidade Federal de Viçosa, Viçosa 36570-900, Brazil; samilalcastro@gmail.com (I.S.L.C.); rejane.lopes@yahoo.com (R.d.L.F.-L.); tallesmaciel@gmail.com (T.E.F.M.); juancarlos.florez2@gmail.com (J.C.F.); eunizemaciel@gmail.com (E.M.Z.); zambolim@ufv.br (L.Z.)
2 BioDiscovery Institute, University of North Texas, Denton, TX 76201, USA; savio.desiqueiraferreira@unt.edu
3 ED&FMan, Carcafe, Bogota 110231, Colombia
4 Brazilian Agricultural Research Corporation (Embrapa), Embrapa Coffee, Brasilia 70770-901, Brazil
* Correspondence: eveline.caixeta@embrapa.br; Tel.: +55-31-3612-2436
† These authors contributed equally to this work.

**Abstract:** Coffee leaf rust is caused by *Hemileia vastatrix* Berk. and Broome and is the most important coffee disease in all regions where coffee is cultivated. Here, we sought to sequence the transcriptome of *H. vastatrix* race XXXIII to obtain a database for use as a reference in studies of the interaction between the fungus and coffee. In addition, we aimed to identify differentially expressed genes that have the potential to act as effector proteins during the interaction. Sequencing of cDNA libraries from uredospores and from compatible and incompatible interactions at different key time points generated about 162 million trimmed reads. We identified 3523 differentially expressed genes. The results suggested that the fungal transcriptome is dynamically altered over the course of infection and that the interaction with a susceptible plant upregulates a larger set of fungal genes than the interaction with a resistant plant. Co-expression network analysis allowed us to identify candidate genes with the same expression pattern as that of other effectors of *H. vastatrix*. Quantitative PCR analysis identified seven transcripts that may be effectors involved in the coffee–*H. vastatrix* interaction. This information provides a basis for obtaining new insights into the molecular mechanisms of infection in this pathosystem. Understanding gene expression during the infection process may contribute to elucidating the molecular mechanisms leading to the breakdown of resistance by new physiological races of the fungus.

**Keywords:** coffee leaf rust; *Coffea* sp.; candidate effector; differentially expressed genes; plant–pathogen interaction

## 1. Introduction

*Hemileia vastatrix* Berk. and Broome causes the most important coffee leaf rust disease in all regions where coffee is cultivated [1,2]. The disease causes leaves to fall prematurely, which results in the death of branches and low photosynthesis rates, ultimately reducing the coffee yield in the following year by 27% to 50% [2,3]. In addition, leaf rust negatively impacts coffee quality, affecting the chemical composition of the beans and the beverage [4]. Between 2008 and 2013, a severe epidemic of the disease was verified in Central and South America, particularly in Nicaragua, Ecuador, El Salvador, Panama, Honduras, Peru, and Colombia [1,2,5]. In Brazil, a major producer, the fungus is widely distributed

in all producing areas of *Coffea arabica* L. and *C. canephora* Pierre ex A. Frohener, causing significant economic losses [3]. The discovery of *H. vastatrix* in Hawaii in late 2020 highlights that this disease now colonizes all major coffee-growing regions of the world [6,7].

The use of resistant cultivars is one of the main strategies for disease control. However, the emergence of new fungal races has broken down resistance in the field [1,8,9]. Different physiological races may express different virulence genes, which may or may not confer the ability to infect a particular coffee genotype. To date, more than 50 races of *H. vastatrix* have been described worldwide [1,10], of which 16 were identified in Brazil: I, II, III, VII, X, XIII, XV, XVI, XVII, XXI, XXII, XXIII, XXIV, XXV or XXXI, XXXIII, and XXXVII [11,12]. The race XXXIII is complex, containing two or three virulence genes ($v_{5,7}$ or $v_{5,7,9}$). In Brazil, it was identified in some cultivars that had been released as resistant to coffee leaf rust, including the cultivar Oeiras MG 6851 [12], which was derived from a cross between *C. arabica* cv. Caturra Vermelho CIFC 19/1 and Híbrido de Timor CIFC 832/1 (a natural hybrid between *C. arabica* and *C. canephora*). Thus, the breakdown of resistance by this fungal race poses a challenge to coffee breeding programs using Híbrido de Timor (HdT). HdT accessions constitute the main source of resistance genes to rust and other diseases in coffee genetic breeding worldwide [3,10,13], raising concerns about this disease.

*H. vastatrix* is a biotrophic fungus that depends entirely on living plant cells for its growth and reproduction. This mode of interaction involves a prolonged and effective suppression of the host immune system and, simultaneously, the induction of specific host genes for establishing biotrophy [14,15]. During the infection process, the fungus develops specialized infective structures called haustoria, which are responsible for the uptake of nutrients, effector delivery, and regulation of the interaction between the fungus and the host [15–18]. Additionally, haustoria must not be recognized by the host to avoid defense reactions [16] and induce structural changes in host cells, such as cytoskeletal rearrangement, nucleus migration, and chromatin condensation [19]. These changes are believed to be induced by the action of effectors produced in the haustoria, which are secreted into the extra-haustorial matrix and translocated into the plant cell. Once within host cells, effectors can alter their metabolism and defense pathways. Plant resistance responses to rusts are usually observed after haustorium formation [20]. Thus, the pathogen secretes effector proteins to escape plant immune responses and colonize the host [21].

The coffee–*H. vastatrix* interaction follows the gene-for-gene relationship [22]. The resistance of coffee plants is governed by at least nine major dominant genes ($S_H1$–$S_H9$) that are single or associated; however, other major and minor genes may also be involved [23–26]. Similarly, the virulence of the fungus is associated with different virulence genes ($v_1$–$v_9$) [1,8].

Differentially expressed genes have been identified in coffee plants infected by the fungus, including genes related to recognition, signaling, cellular communication, defense, and metabolic processes [27]. In contrast, little information was available for the fungus until recently, with few genomic sequences deposited in public databases. This is probably related to the difficulty in separating transcripts from the plant and fungus, especially in the absence of complete genomic sequences, which is the major challenge in plant–pathogen interaction studies. Some transcriptomic studies [28,29] and the partial sequencing of the *H. vastatrix* genome [30,31] have provided new information, especially on fungal genes involved in the signaling, establishment, and maintenance of biotrophy. However, information on the XXXIII race of *H. vastatrix* is scarce, and relatively little progress has been made in understanding the molecular mechanisms of *H. vastatrix* infection.

Here, we sought to sequence and assemble the transcriptome of *H. vastatrix* race XXXIII to obtain a reference database for the study of the interaction between coffee and this fungal race. In addition, we evaluated the transcriptomes of infected susceptible and resistant coffee plants, generating information on differentially expressed fungal genes at different stages of the infection process in both compatible and incompatible interactions.

These genes have the potential to act as effector proteins. Our findings provide new insights into the molecular bases of the infection process in this pathosystem.

## 2. Materials and Methods

### 2.1. Cytological Evaluation of the Infection Process and Plant Responses

Cytological analysis was performed in order to select the key time points of coffee–*H. vastatrix* interaction to develop the fungus transcriptome. This analysis was previously published in detail by Freitas et al. [32]. In this work, plants of *Coffea arabica* cv. Caturra CIFC 19/1 ($S_H5$) and Híbrido de Timor CIFC 832/1 ($S_H5, 6, 7, 8, 9$?) were inoculated with fresh uredospores of *H. vastatrix* race XXXIII ($v_{5,7}$ or $v_{5,7,9}$) to establish a compatible and an incompatible interaction, respectively. Pre-penetration fungal growth stages (germinated uredospores and appressoria formation over stomata) were visualized on leaf pieces with a light microscope as previously described [33]. For time-course studies of fungal growth and plant cell responses, cross sections of infected leaf fragments made with a freezing microtome were submitted to blue lactophenol staining and epifluorescence test [34–36]. Autofluorescence is thought to indicate the presence of phenolic-like compounds, and cytoplasmic autofluorescence and/or browning is frequently associated with plant cell death [37,38]. Observations were made with an Olympus BX-41 microscope equipped with an HBO 100 W mercury bulb (UV light, excitation filter 330–385 nm). Evaluations were performed at different times: 10, 17, 24, 48, 72, and 96 h post-inoculation (hpi); and 17- and 21-days post-inoculation (dpi).

### 2.2. Sequencing of H. vastatrix and Coffee-Fungus Interaction Libraries

Fresh uredospores of *H. vastatrix* race XXXIII were suspended in sterile distilled water containing 0.02% Tween (*v/v*) and homogeneously distributed in polystyrene plates containing a thin layer of distilled water. The plates were maintained at 22 °C in the dark for 2 and 12 h to obtain hydrated uredospores and germinated uredospores, respectively. After this period, the material was collected and centrifuged at 18,000× *g* for 5 min. The supernatant was discarded and the pellet containing the spores was immediately frozen in liquid nitrogen and stored at −80 °C. Total RNA was extracted with a RNeasy Plant Mini Kit (Qiagen) following the manufacturer's recommendations. RNA concentration and integrity were monitored using the Quant-iT™ RiboGreen®® RNA Reagent (Invitrogen, Inchinnan Business, Park Paisley, UK) and RNA 6000 Nano Kit (Agilent Technologies, Waldbronn, Germany), respectively. cDNA synthesis and library preparation were performed at Genomics Laboratory (North Carolina State University (NCSU), Raleigh, NC, USA) using the Mint-2 cDNA Synthesis (Evrogen, Moscow, Russia), Trimmer–2 cDNA normalization (Evrogen), and TruSeq DNA Sample Preparation (Illumina, San Diego, CA, USA) kits. The cDNA libraries called "HU" (hydrated uredospores) and "GU" (germinated uredospores) were sequenced by Illumina MiSeq, generating 250 bp paired-end reads.

Coffee–*H. vastatrix* interaction libraries were constructed as described previously [27]. Briefly, plants of *C. arabica* cv. Caturra CIFC 19/1 (genotype $S_H5$) and plants of Híbrido de Timor CIFC 832/1 (genotype $S_H5, 6, 7, 8, 9$?) were inoculated with fresh uredospores of *H. vastatrix* race XXXIII (genotype $v_{5,7}$ or $v_{5,7,9}$) to establish a compatible and an incompatible interaction, respectively. In each plant, four leaves were inoculated, and one non-inoculated leaf was used as a control. Based on our previous cytological evaluation (Section 2.1), the inoculated leaves were collected at 12, 24, and 96 hpi, and at 17 dpi. After collection, the leaves were immediately frozen in liquid nitrogen and maintained at −80 °C for RNA extraction. Each sample consisted of a pool comprising three leaves from three different plants. All further steps, including RNA extraction, cDNA library preparation, and sequencing were performed as described above.

*2.3. Read Processing, Mapping, Transcriptome Assembly, Expression Quantification, and Identification of Differentially Expressed Genes (DEGs)*

All bioinformatic steps are summarized in Supplementary Figure S1. Raw data quality was analyzed using the FastQC software (http://www.bioinformatics.babraham.ac.uk/projects/fastqc/ (accessed on 3 August 2016), and low-quality reads and adaptor sequences were eliminated with Trimmomatic software [39]. Reads originated from coffee plants were filtered by mapping against the *C. canephora* reference genome (http://coffee-genome.org/coffeacanephora (accessed on 8 August 2016) with Tophat2 [40]. Unmapped reads against *C. canephora* were extracted from alignment output files (unmapped.bam) using SAMtools fastq and then repaired with the Pairfq makepairs command (https://github.com/sestaton/Pairfq/ (accessed on 16 August 2016). These reads were mapped against the *H. vastatrix* race XXXIII reference genome [31] with Tophat2 software.

Reads from all libraries (HU, GU, and coffee–*H. vastatrix* interactions) mapped against the *H. vastatrix* reference genome were used in the transcriptome assembly using Cufflinks software [39], and then merged with Cuffmerge, which is part of the Cufflinks package.

The assembled transcriptome was used as a reference for expression quantification using the Kallisto software [40]. The read count table generated by this analysis was used in DESeq [41] and EdgeR [42], and only DEGs detected by both packages were considered. An expression heatmap was built with Gene Cluster 3.0 (http://bonsai.hgc.jp/~mdehoon/software/cluster/ (accessed on 15 September 2016) and TreeView 1.1 (http://jtreeview.sourceforge.net (accessed on 16 September 2016)) using transcript per million (TPM) $\log_2$ expression values.

*2.4. Co-Expression Analysis*

TPM log2 expression values were used for co-expression analysis with the WGCNA R-package [43]. The characterized *H. vastatrix* race II effector HvEC-016 [44] was used to find its putative homologue in race XXXIII by Blastn. Then, we searched in the co-expression modules to find which one comprises the HvEC-016 homologue, aiming to identify other genes with the same expression pattern, and filtering them by putative effector proteins as described above. The network was visualized in Cytoscape [45].

*2.5. Functional Annotation*

Differentially expressed genes were annotated using BlastN and BlastX tools against several databases, including NCBI-NR, Pathogen Host Interactions (PHI-base) [46], PFAM [47], Swissprot [48], mRNA-RefSeq (NCBI), and TransposonPSI (http://transposonpsi.sourceforge.net/ (accessed on 18 October 2016). Additionally, Blast2GO (https://www.blast2go.com/ (accessed on 18 October 2016) was used for gene ontology annotation. In order to identify putatively secreted proteins, TargetP [49], SignalP [50], and TMHMM [51] software was used to identify putative subcellular localization, signal peptides involved in secretion, and transmembrane regions, respectively. The classification as putative secreted protein was based on the classification by all analyses, including the classification as secreted by TargetP, the presence of a signal peptide for secretion detected by SignalP, and the absence of transmembrane regions detected by TMHMM [52].

*2.6. Differentially Expressed Gene Validation by Real-Time qPCR*

We selected 11 DEGs for qPCR validation, focusing on early stages of the interaction (12 and 24 hpi). A new set of plants from the same susceptible and resistant genotypes were inoculated with fresh uredospores of *H. vastatrix* race XXXIII. Samples were collected at 12 and 24 hpi and used for RNA extraction as described in Section 2.2. As a control, hydrated and germinated uredospores were also harvested and used for RNA extraction. Total RNA was treated with DNAse RQ1 (Promega), and the cDNA was synthetized with

Improm-II kit (Promega) following the manufacturer's instructions. Real-time PCR reactions were carried out using GoTaq qPCR Master Mix (Promega) in the 7500 Real-Time PCR System (Applied Biosystems, Waltham, MA, USA) with default reaction cycling parameters. Beta-tubulin and cytochrome C oxidase subunit III [46] were used as endogenous controls, and relative expression was calculated as previously described [47].

## 3. Results

### 3.1. Transcriptome of H. vastatrix

The transcriptome was obtained based on libraries from hydrated and germinated uredospores of *H. vastatrix* (race XXXIII) and from compatible (coffee *cv* Caturra CIFC 19/1 and race XXXIII) and incompatible (coffee Híbrido de Timor CIFC 832/1 and race XXXIII) interactions at different key time points. The time points were selected by the cytological evaluation of the infection process [32]. Based on these data, the two coffee genotypes (resistant and susceptible) could be readily differentiated by their cytological responses in early events of fungal infection (Table 1, Figure 1). Initial responses occurred particularly in the stomatal cells of both genotypes at 17 hpi and consisted of a hypersensitive response (HR) and the accumulation of phenolic-like compounds (Figure 1C). In the resistant coffee genotype (Híbrido de Timor CIFC 832/1), these responses were observed in 18% of the infection sites at 17 hpi, reaching 65% and 93% at 24 and 96 hpi, respectively (Table 1). The fungus stopped growth at the stage of penetration hypha in the majority (65%) of the infection sites at 96 hpi (Figure 1A). In the susceptible leaves (Caturra CIFC 19/1), HR responses were observed in about 30% of the infection sites from 17 to 96 hpi (Table 1). Although these values are higher than those reported in other compatible coffee–*H. vastatrix* interactions [35,48], the fungus was able to grow and colonize the host tissues, producing a large number of haustoria and intercellular hyphae, and sporulating around 17 dpi (Figure 1B). Based on these data, the libraries for transcriptome analysis were constructed in 12, 24, and 96 hpi, and 17 dpi.

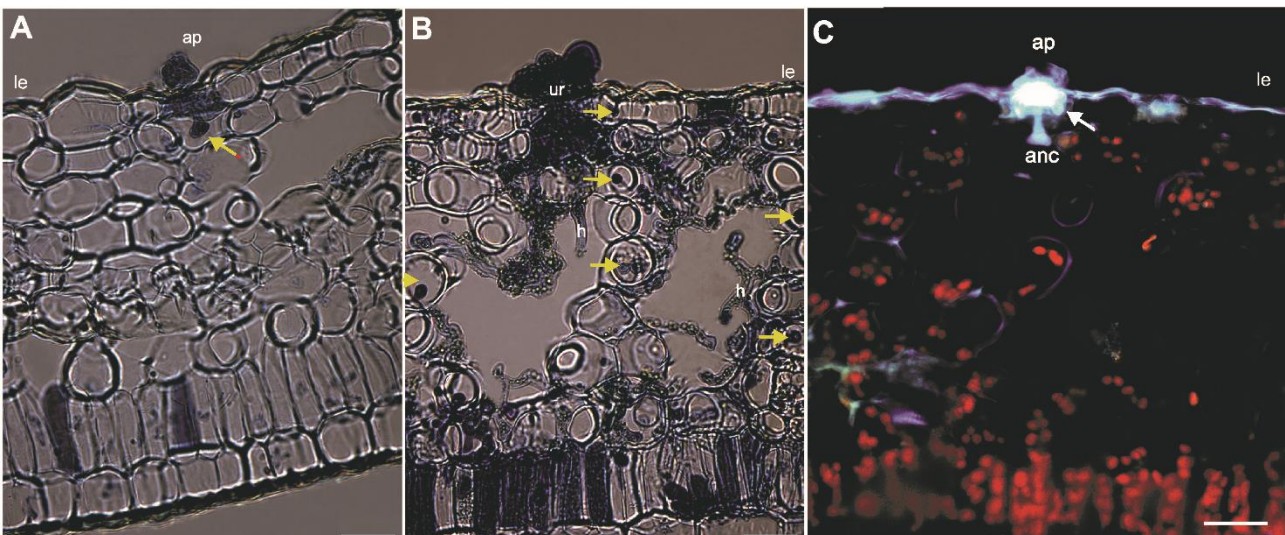

**Figure 1.** Colonization of leaf tissues by race XXXIII of *Hemileia vastatrix*. (**A**,**B**) showing cotton blue lactophenol staining in a resistant plant at 96 hpi (**A**) and a susceptible plant at 21 dpi (**B**). (**A**) Appressorium (ap) over stomata and a penetration hypha into the substomatal chamber (arrow). (**B**) Uredospore (ur), hyphae (h), and haustoria (arrows) within the cells of the lower epidermis (le) and mesophyll. (**C**) Cytological responses induced by the fungus in the resistant genotype at 24 hpi. Epifluorescence test (UV light): an anchor (anc) associated with autofluorescence of guard and subsidiary cells (arrow white solid) indicating plant cell death through the hypersensitive response (HR) and accumulation of phenolic-like compounds. Bar = 20 μm. This figure was taken from Freitas et al. [32].

**Table 1.** Percentage of infection sites with hypersensitive cell death and accumulation of phenolic-like compounds (monitored by the presence of autofluorescent and/or browning cells) in leaves of the coffee genotypes resistant and susceptible to *Hemileia. vastatrix* (race XXXIII).

| Hours Post-Inoculation | % of Infection Sites with Autofluorescent and/or Browning Cells in Leaves of the Coffee Genotypes | |
|---|---|---|
| | Híbrido de Timor CIFC 832/1 (R) | Caturra CIFC 19/1 (S) |
| 10 | 0 | 0 |
| 17 | 18 ± 3 | 30 ± 5 |
| 24 | 65 ± 13 | 30 ± 5 |
| 48 | 73 ± 8 | 30 ± 5 |
| 72 | 83 ± 8 | 28 ± 6 |
| 96 | 93 ± 3 | 28 ± 3 |

R = resistant genotype; S = susceptible genotype; values are mean ± standard deviation, data recorded from 60 infection sites in each time point were presented as the combined values of two independent experiments because no significant differences were found between them. This table was taken from Freitas et al. [32].

Sequencing of all cDNA libraries generated about 162 million trimmed reads (Table 2). As the interaction libraries comprised transcripts from both organisms, we first filtered plant reads by mapping all reads against the coffee reference genome [49]. About 95 million reads mapped to the coffee genome and were discarded for further analysis (Table 2). A very small number of reads (~1%) from *H. vastatrix*-exclusive (uredospores) libraries (HU and GU) mapped to the plant genome (Table 2). These may represent reads from genes conserved across kingdoms or even horizontal gene transfer; however, further investigation is needed to verify these hypotheses. From the remaining 67 million reads, over 24 million reads mapped against the *H. vastatrix* race XXXIII reference genome [31], while 43 million reads did not map to the coffee or *H. vastatrix* reference genomes. We believe that these unmapped reads may be gaps within the analyzed reference genomes of coffee and *H. vastatrix*.

The reads mapped against the *H. vastatrix* genome were assembled to produce the *H. vastatrix* transcriptome. The assembly resulted in 43.6 Mb, comprising a total of 29,812 transcripts with an average length of 1463 nucleotides. The longest and shortest transcript had 11,657 and 51 nucleotides, respectively.

**Table 2.** Overall number of sequencing and mapping reads against coffee and *Hemileia vastatrix* reference genomes.

| Libraries | Raw Reads | Trimmed Reads | Mapped Reads against Coffee | Unmapped Reads against Coffee | *H. vastatrix* Reads * |
|---|---|---|---|---|---|
| **Uredospores** | | | | | |
| Hydrated (HU) | 19,095,474 | 15,120,725 | 224,132 | 14,896,593 | 10,685,403 |
| Germinated (GU) | 11,854,206 | 9,780,528 | 111,336 | 9,669,192 | 6,116,869 |
| **CIFC 19/1 (S)** | | | | | |
| 12 h | 24,937,788 | 21,535,680 | 14,662,195 | 6,873,485 | 1,040,374 |
| 24 h | 19,149,370 | 15,285,996 | 9,698,668 | 5,587,328 | 552,489 |
| 96 h | 13,416,692 | 10,677,026 | 6,380,984 | 4,296,042 | 397,579 |
| 17 d | 27,295,036 | 20,334,878 | 10,244,133 | 10,090,745 | 3,912,843 |
| **CIFC 832/1 (R)** | | | | | |
| 12 h | 22,328,618 | 19,101,578 | 14,698,398 | 4,403,180 | 666,275 |
| 24 h | 31,061,884 | 24,885,905 | 20,409,662 | 4,476,243 | 648,268 |
| 96 h | 12,359,240 | 9,438,008 | 6,460,518 | 2,977,490 | 172,604 |
| 17 d | 18,684,474 | 16,204,815 | 12,188,572 | 4,016,243 | 140,861 |
| Total | 200,182,782 | 162,365,139 | 95,078,598 | 67,286,541 | 24,333,565 |

* Unmapped reads against coffee genome and mapped against *H. vastatrix* genome; S = susceptible genotype; R = resistant genotype.

Considering all mapped reads (Table 2), there was a reduction in mapped reads against the *H. vastatrix* reference genome along the time course in the incompatible interaction, starting with ~4% at 12 hpi and decreasing to 1% at 17 dpi (Figure 2). The opposite was observed in the compatible interaction, where the number of reads mapped to the fungal genome increased from ~6% at 12 hpi to ~27% at 17 dpi (Figure 2).

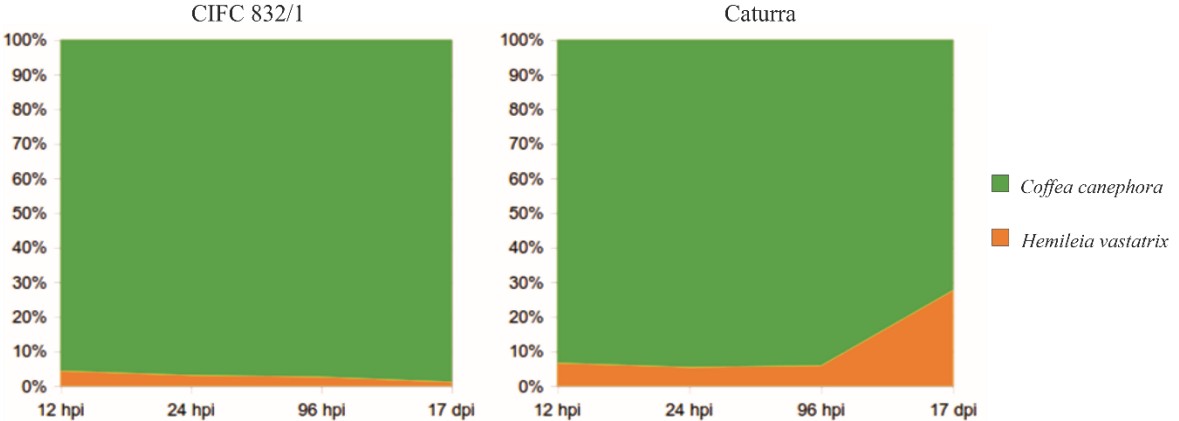

**Figure 2.** Percentage of all mapped reads from each interaction against the two reference genomes (*Coffee canephora* genome and *Hemileia vastatrix* genome). Resistant genotype CIFC 832/1 (incompatible interaction); susceptible genotype Caturra CIFC 19/1 (compatible interaction); hpi, hours post-inoculation; dpi, days post-inoculation.

### 3.2. Changes in the Fungal Transcriptome when Interacting with the Plant

The assembled transcriptome of *H. vastatrix* (race XXXIII) allowed us to carry out gene expression analysis and quantification by using principal component analysis (PCoA, Figure 3A) and hierarchical clustering (Figure 3B). Both analyses showed that the gene expression pattern of the uredospore libraries (hydrated (HU) and germinated (GU)) was similar (Figure 3). However, it completely differed from the interaction libraries, as they were clustered separately from all other samples. Although the GU library consisted of uredospores incubated for 12 h in petri dishes containing water, which is the same time used for the first sample of the interaction libraries (12 hpi), the expression pattern in both libraries was completely different. This suggests that the fungus can readily recognize the plant and that this interaction is crucial for changes in the fungal transcriptome.

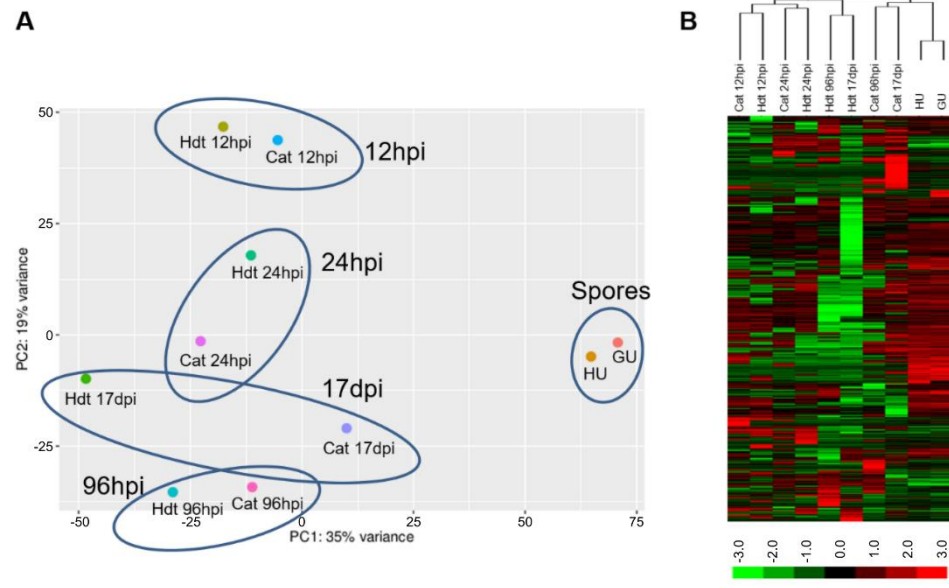

**Figure 3.** Principal component analysis (PCoA) and hierarchical clustering. PCoA (**A**) and hierarchical clustering (**B**) of *Hemileia vastatrix* gene expression. Cat, Caturra (susceptible genotype); Hdt, Híbrido de Timor (resistant genotype); HU, hydrated uredospores; GU, germinated uredospores. In (**B**), each column represents the fold change in transcript levels ($log_2$ of TPM). Upregulated and downregulated genes are shown in red and green, respectively. The intensity color scale indicates the level of expression.

On the other hand, the overall gene expression of the interaction libraries was very similar at the initial stages of infection, regardless of the plant genotype. At 12 and 24 hpi, both susceptible and resistant genotypes were clustered together (Figure 3A,B). As the infection progressed, however, the differences between compatible and incompatible interactions became clearer. The libraries of 96 hpi were kept in different clusters in the heatmap (Figure 3B), and the 17 dpi libraries were separated by both analyses (Figure 3A,B). Interestingly, the compatible interaction libraries (Cat) at 96 hpi and 17 dpi were closer to uredospore libraries (Figure 3B). Since the infection in these samples was successful, the finding may suggest a shifting of the transcriptome to sporulation phase.

*3.3. Differentially Expressed Genes (DEGs)*

Library comparisons were carried out to identify differentially expressed genes (DEGs) that could explain the fungus transcriptional changes required to establish a successful biotrophic interaction (Table 3). First, we used a pool of uredospore libraries (GU and HU) as a control (reference) and analyzed the DEGs in a time course for each interaction. Then, we compared each time point individually by contrasting the compatible to incompatible interaction (reference) to identify differences in the fungal transcriptome at a same time point when facing an effective (CIFC 832/1, HdT) or ineffective (Caturra CIFC 19/1, Cat) defense. These approaches yielded a total of 3523 DEG occurrences, representing 2040 DEGs after excluding genes that appeared in more than one comparison (Table 3).

**Table 3.** Number of DEGs in all comparisons.

| Experiment | Reference (Control) | Upregulated | Downregulated | Total |
|---|---|---|---|---|
| HdT (Resistant) 12 hpi | Spores [a] | 170 | 168 | 338 |
| HdT (Resistant) 24 hpi | Spores | 157 | 133 | 290 |
| HdT (Resistant) 96 hpi | Spores | 69 | 45 | 114 |
| HdT (Resistant) 17 dpi | Spores | 24 | 13 | 37 |
| Cat (Susceptible) 12 hpi | Spores | 248 | 203 | 451 |
| Cat (Susceptible) 24 hpi | Spores | 227 | 184 | 411 |
| Cat (Susceptible) 96 hpi | Spores | 179 | 112 | 291 |
| Cat (Susceptible) 17 dpi | Spores | 312 | 236 | 548 |
| Cat (Susceptible) 12 hpi | HdT (Resistant) 12 hpi | 263 | 154 | 417 |
| Cat (Susceptible) 24 hpi | HdT (Resistant) 24 hpi | 178 | 208 | 386 |
| Cat (Susceptible) 96 hpi | HdT (Resistant) 96 hpi | 144 | 86 | 230 |
| Cat (Susceptible) 17 dpi | HdT (Resistant) 17 dpi | 0 | 10 | 10 |
| Total | | 1971 | 1552 | 3523 |

[a] Pool of hydrated (HU) and germinated (GU) uredospores libraries; Cat = Caturra; HdT = Híbrido de Timor.

In the first approach (using uredospores as a reference and analyzing each type of interaction individually), we were able to identify genes involved in the attempt of the fungus to infect the plant in both susceptible and resistant genotypes. The number of DEGs at 17 dpi was 37 and 548 in the incompatible and compatible interactions, respectively, which reflects the number of fungal reads in these samples (Figure 2).

The Venn diagram analysis (Figure 4) showed that most of the up- and downregulated genes were exclusive to each time point for both compatible (Figure 4A,B) and incompatible (Figure 4C,D) interactions. Only a few genes were differentially expressed at all time points, with 16 up- and seven downregulated genes in the compatible interaction, and two up- and five downregulated genes in the incompatible interaction. A high number of DEGs were exclusive to the compatible interaction at 17 dpi (202 up- and 163 downregulated genes), which suggests a differential transcriptome profile at late stages of the infection, as demonstrated in Figure 3B.

A comparison of the upregulated genes between compatible and incompatible interactions at early stages of infection revealed that the number of exclusive genes was two times greater in the compatible than in the incompatible interaction at 12 hpi (Figure 4E) and 24 hpi (Figure 4F).

The second approach used to identify DEGs consisted of comparing the compatible and incompatible interactions samples at a same time point, resulting in four comparisons (Table 3). The number of DEGs ranged from 10 (at 17 dpi) to 417 (at 12 hpi). Venn diagrams of these analyses (Figure 5) showed that different time points shared low numbers of up- (Figure 5A) or downregulated (Figure 5B) genes.

Functional annotation was carried out using blast search against NR, RefSeq mRNA, PHI, TransposonPSI, Swissprot, and PFAM databases (Supplementary Table S1). The first hit in each database was manually checked to determine the best annotation for each DEG, which was then separated by functional categories, followed by an in-house pipeline to identify putative secreted proteins (see methods Section 2.4). The species with more matches against the *H. vastatrix* transcriptome in NCBI-NR (Supplementary Figure S2) were *Melampsora larici-populina* (619), *Puccinia graminis* (253), *P. triticina* (247), and *P. striiformis* (228). There were only ten matches with *H. vastatrix* transcripts.

In the PHI database (Supplementary Figure S2), 614 DEGs had hits against 50 different species, and *Fusarium graminearum* was the species with more hits (140). On the other hand, 393 transcripts, from which 64 are putatively secreted, had no hits against any species and may represent *H. vastatrix*-specific transcripts.

An overview of functional categories of the DEGs is shown in Figure 6. By analyzing all DEGs (Figure 6A), we found that 718 (~35%) have unknown function, of which 87 are putatively secreted. The second most representative category was "signaling" (including G-proteins, receptor kinases, kinases, and transcription factors), followed by "protein metabolism" (comprising proteases and translation-related proteins), and "transport" (including membrane transporters).

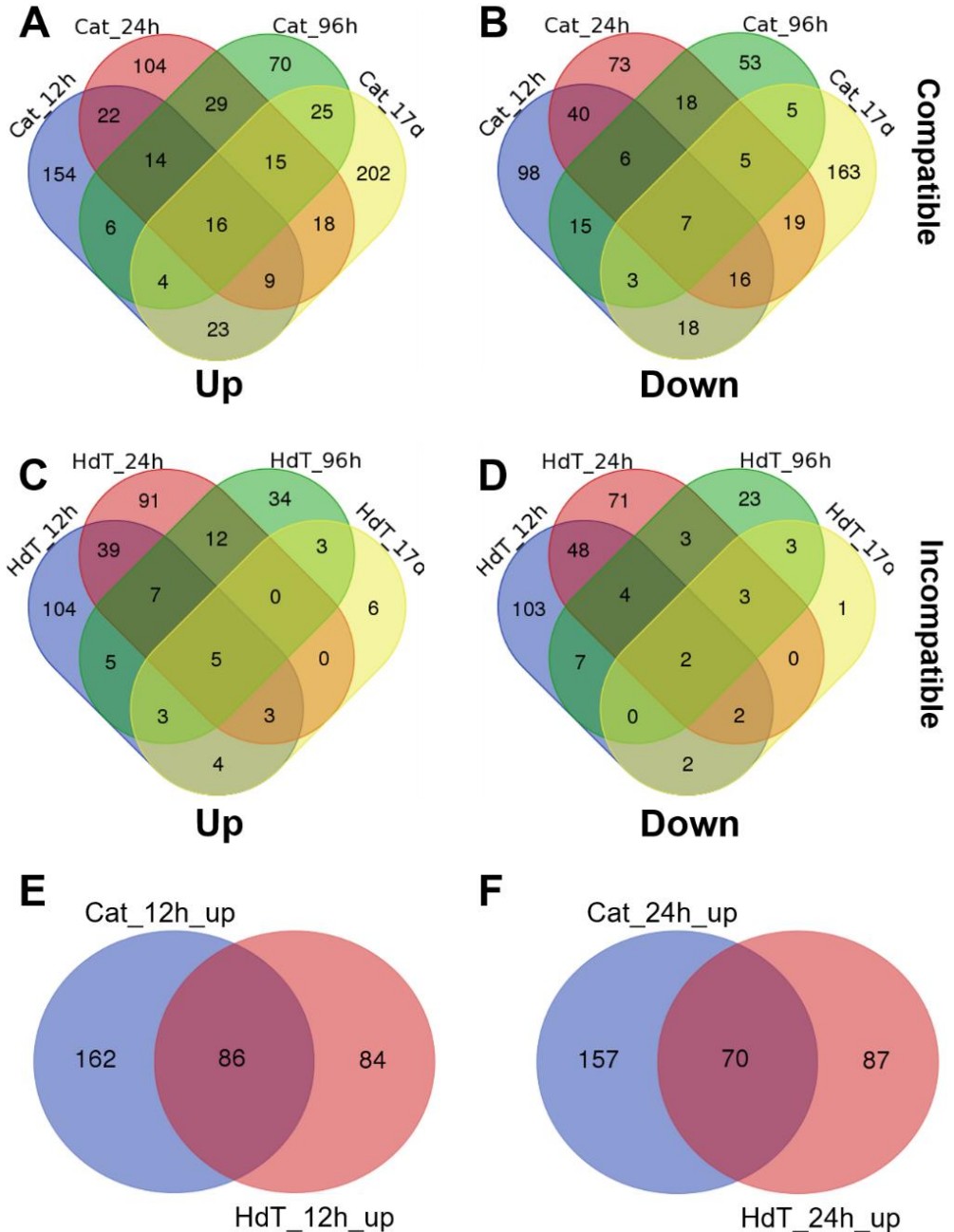

**Figure 4.** Venn diagrams of DEGs with compatible and incompatible interactions compared to ure-dospores in a time course manner. Up- (**A**) and down- (**B**) regulated genes in compatible interactions (Cat, Caturra); Up- (**C**) and down- (**D**) regulated genes in incompatible interactions (HdT, Híbrido de Timor). Comparison of the upregulated genes between compatible and incompatible interactions at 12 (**E**) and 24 (**F**) hpi. Time points: 12 hpi, 24 hpi, 96 hpi, and 17 dpi.

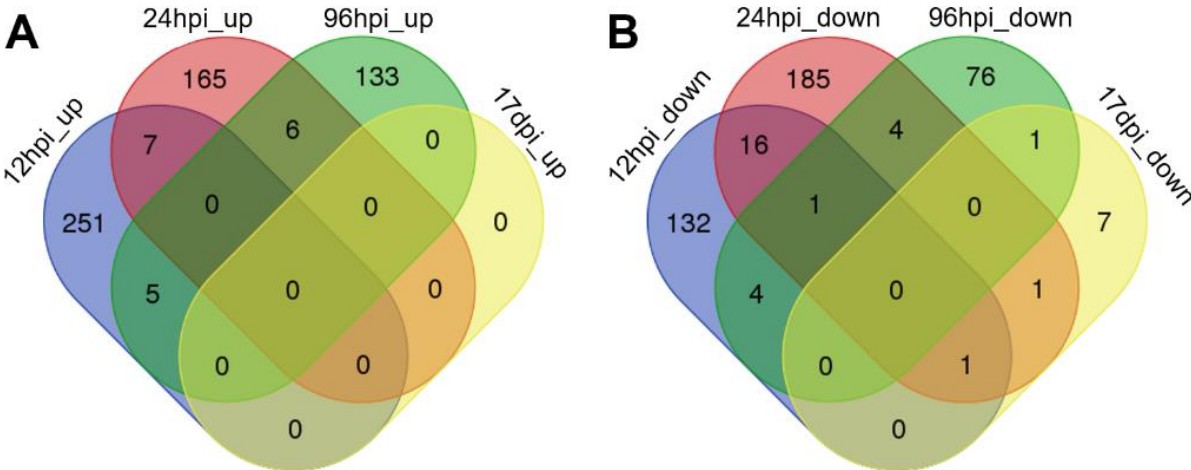

**Figure 5.** Venn diagrams comparing DEGs between compatible and incompatible interactions at each time point. (**A**) Upregulated genes; (**B**) downregulated genes.

**Figure 6.** Functional categories of all DEGs identified. Putative secreted proteins are shown for each category. (**A**) All DEGs; (**B**) incompatible interaction; (**C**) compatible interaction; (**D**) compatible versus incompatible interactions.

To get a better understanding of the functional alterations based on the transcriptome, we filtered the functional categories by each type of DEG identification approach. In the first strategy (analyzing each type of interaction separately), the main functional categories in the incompatible interaction were "unknown function," "signaling," "protein metabolism," and "RNA metabolism" (Figure 6B). In the compatible interaction, the "unknown function," "signaling," "transport," and "protein metabolism" categories were predominant (Figure 6C). Both interactions shared a similar pattern, including "unknown function," "signaling," and "protein metabolism" categories. On the other hand, the "RNA metabolism" category, present only in the incompatible interaction, might reflect the pathogen attempt to alter its gene expression to counterattack the plant defense. In the second approach (comparing each time point from both interactions), the same pattern could be identified, with "unknown function," "signaling," "protein metabolism," and "RNA metabolism" as the main categories (Figure 6D). In this analysis, the percentage of putative secreted proteins was much smaller than that in other analyses, suggesting that the putative secreted effectors, which play important roles in virulence and avirulence, do not differ significantly between interactions.

### 3.4. Identification of Putative Effector Homologues from Other Races and Co-Expression Network Analysis

To identify putative pathogen effectors, we carried out a co-expression network analysis to find candidate genes with the same expression pattern as other *H. vastatrix* effectors. Genes with an expression pattern similar to that of known effectors also have a high chance of being effectors. Thus, the effectors need to be characterized and used as bait to find others with similar expression patterns. Here, we compared the sequence of 100 previously reported putative effectors from *H. vastatrix* races II and XIV [28–30,44] with that of the race XXXIII transcriptome (Figure 7). Forty percent of the previously reported putative effectors showed 100% identity with a transcript from race XXXIII, while 47% shared 83% to 99% identity. In addition, six putative effectors were not covered entirely (coverage < 90%) and seven did not match any transcript from race XXXIII (no hit), suggesting a significant sequence variability in putative effector homologues among these races.

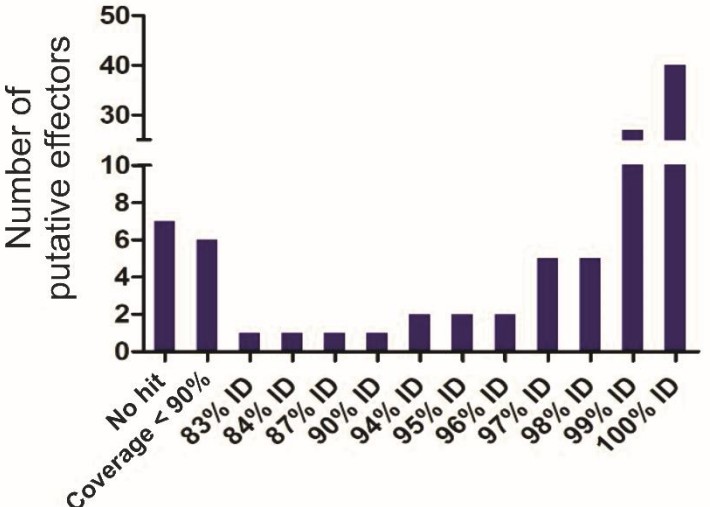

**Figure 7.** Sequence variation among putative effectors previously reported in the races II and XIV [28,30,53] when compared to the transcriptome sequence of race XXXIII (this study). Identical (100% ID) proteins could be found for 40 out of 100 proteins, and 47 shared 83% to 99% identity. Seven proteins had no hit against race XXXIII transcriptome, and six proteins showed coverage lower than 90% by a transcript from race XXXIII.

This dataset includes the recently characterized effector HvEC-016 from *H. vastatrix* race II, which would be the avirulence gene corresponding to $S_{H}1$ [44] and thus, it is a good bait for co-expression analysis. Transcript 2156 is the homologue of HvEC-016 in race XXXIII and the comparison of their sequences showed mutations and INDELs in the 3'-end of the open reading frame (ORF) that completely changes the C-term of the protein (Figure 8A,B). However, cysteine residues that may be important for native folding (by disulfide bridges) of the effector are still present (Figure 8B), and both nucleotide and protein alignment showed > 95% identity (Figure 8C).

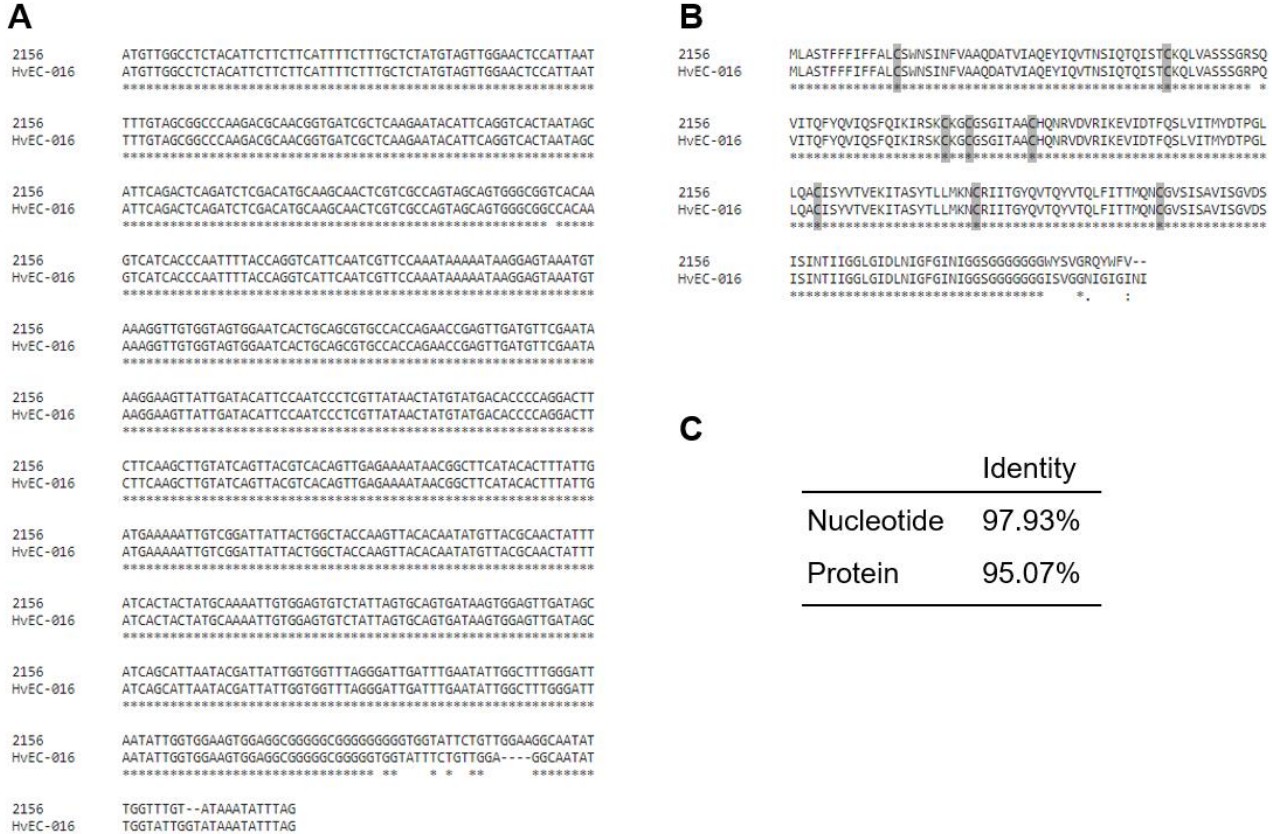

**Figure 8.** Alignment comparing *Hemileia vastatrix Avr1* from race II, HvEC-016 [44], and its putative homologue in race XXXIII (2156, this study). (**A**) nucleotide alignment; (**B**) protein alignment with cysteine residues shadowed in gray; (**C**) percent identity between 2156 and HvEC-016. Alignments were carried out by the clustal omega web tool (https://www.ebi.ac.uk/Tools/msa/clustalo/ (accessed on 10 January 2017). In A and B, below the sequences is a key denoting conserved sequence (*) and gap (-). In B, differences between the two protein sequences are denoted as conservative (:), semi-conservative (.), and non-conservative ( ).

After identifying the bait (transcript 2156, the HvEC-016 homologue), WGCNA [43] was used to build a co-expression network with all expression data. The network module on which transcript 2156 is present comprises 212 transcripts (Supplementary Table S2), represented mostly by "unknown" and "protein metabolism" functional categories (Figure 9). In the "protein metabolism" category, 28 of 30 genes were ribosome-related (Supplementary Table S2). Since germination and infection require intense protein synthesis, the result suggested that genes co-expressed with the putative effector are involved in basic cell functions related to growth. On the other hand, over 50% of the co-expressed genes belonged to the "unknown" category, of which 24 are putatively secreted and, therefore, good candidate effectors. The results also showed that, except for the HvEC-016 homologue, none of the 100 putative effectors reported previously are present in the co-

expression module, which may indicate that effector expression diversity among races could be greater than expected.

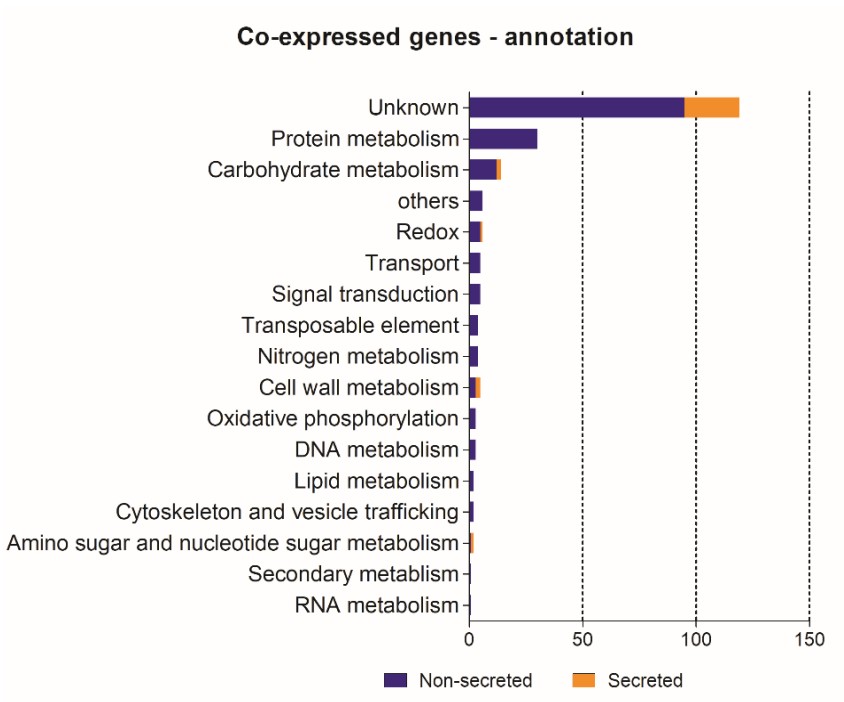

**Figure 9.** Functional categories annotation of transcripts co-expressed with 2156, the putative HvEC-016 race XXXIII homologue. Putative secreted proteins are shown for each category.

## 4. Expression Analysis by qPCR

After identifying DEGs and genes co-expressed with an effector, qPCR was used to validate the results by searching for genes that might be good candidate effectors based on their expression pattern. We focused on unknown and secreted proteins, since the most effectors show unknown functions [50], including *H. vastatrix*-specific transcripts and those with hits against other species. Considering that initial stages are crucial for the establishment of biotrophy and that a very low number of reads could be detected at later stages in the incompatible interaction, the analyses were performed using data from 12 and 24 hpi.

Twelve genes were selected for qPCR analysis (Figure 10), of which 11 were from the DEG list and the transcript 2156. Three transcripts, 5592, 10,268, and 17,877 (Figure 10B–D), which were upregulated at 24 hpi in both interactions, were found to be co-expressed with 2156 (Supplementary Table S2), as confirmed by their similar expression pattern observed by qPCR. The transcripts 5592, 10,268, 17,877, 5591, 7607, 10,732, and 7609 (Figure 10B–H) validated the differential expression detected by RNAseq analysis (Supplementary Table S1). The similar pattern of RNAseq and qPCR was not observed for the remaining transcripts (Figure 10I–L), probably due to the high difference between biological samples within each data point, leading to a high standard error. Since the main objective was to identify possible candidate effectors, we focused on those genes upregulated at early stages of infection (12 or 24 hpi) with a pattern similar to that of transcript 2156. Thus, the transcripts 5592, 10,268, 17,877, 5591, 7607, 10,732, and 7609 (Figure 10B–H) were good candidates for further functional characterization, as they were remarkably upregulated at 12 and 24 hpi. On the other hand, no significant difference in the expression of these transcripts was observed between compatible and incompatible interactions.

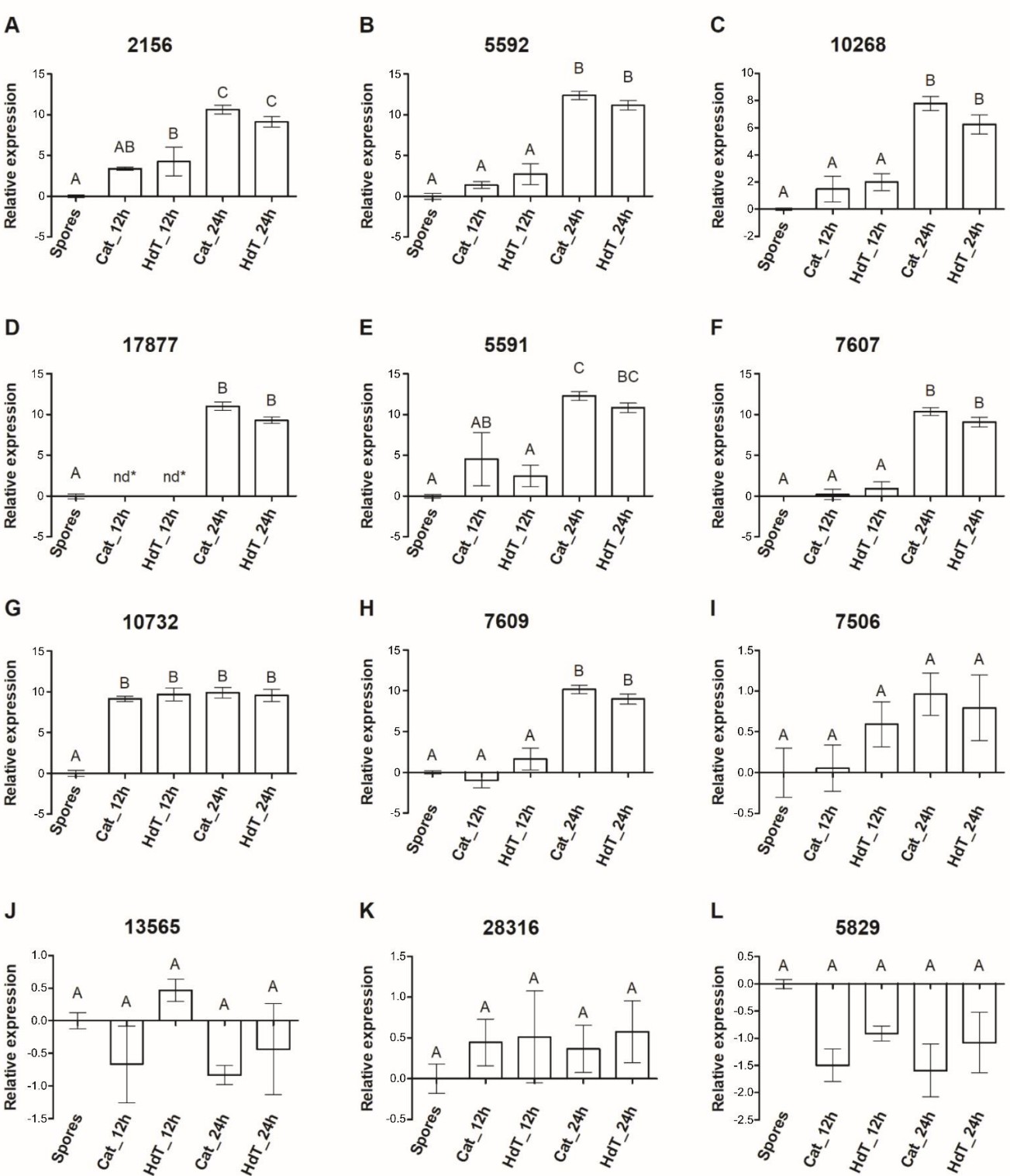

**Figure 10.** qPCR gene expression analysis of selected DEGs and co-expressed genes. Results are shown as log₂ of relative expression, normalized by the expression in spores (log₂ relative expression in spores = 0). N = three biological replicates; error bar = SEM. Different letters at the top of each bar indicate statistically significantly different mean expression values as determined by one-way analysis of variance (ANOVA) followed by Tukey's test (*p*-value < 0.05). Cat, coffee genotype Caturra, compatible interaction; HdT, coffee genotype Híbrido de Timor CIFC832/1, incompatible interaction. (**A–L**) show the expression of each gene, identified by transcript ID number at the top of each graph.

### 5. Discussion

To better understand the interaction between coffee plants and the race XXXIII of *H. vastatrix*, we evaluated fungal growth and plant defense responses in two coffee genotypes, resistant Híbrido de Timor CIFC832/1 and susceptible cv. Caturra, by cytological analysis. The race XXXIII was identified in Brazil infecting some cultivars that had been released as resistant to coffee leaf rust [12], as well as some important resistance sources used in coffee breeding programs, such as Catimor accessions [1].

The major differences in the cytological responses between compatible and incompatible interactions occurred at a late stage of the infection process (17 dpi). This is possibly due to the increased biomass of the fungus in the susceptible plant tissues, since the intensive growth of intercellular hyphae and haustoria culminate in the formation of a dense mycelium taking the whole mesophyll [51]. The HR was initiated at 17 hpi in both interactions but reached 93% of the infection sites at 96 hpi in the resistant genotype and was maintained at 30% in the susceptible genotype. Typically, the HR is associated with race-specific resistance involving the gene-for-gene interactions [52,53]. Nevertheless, and as verified here, several studies have concluded that the accumulation of phenolic compounds may be associated with cell death in host resistance, thus being one of the first lines of plant defense against infection [54,55].

The cytological evaluation also showed that, in susceptible leaves, the fungus pursues its growth in most of the infection sites with an increase in the formation of intercellular hyphae and haustoria in the cells of the spongy and palisade parenchyma, and even of the upper epidermis. In resistant genotypes, fungal growth inside leaf tissues is stopped at different stages of the infection, with higher frequency in the stage of penetration hypha. Taken together, these results demonstrated that Híbrido de Timor CIFC 832/1 halts fungal growth before haustoria differentiation, which indicates a pre-haustorial resistance that is complementary to the post-haustorial resistance generally described for coffee–*H. vastatrix* interactions [34]. For example, the cultivar Tupi shows resistance after the differentiation of secondary haustoria, since $H_2O_2$ burst has been detected at 39 hpi [51]. In addition, our findings are in agreement with a study showing the upregulation of a broad range of genes, several of them resistance-related, in early events of infection (12 and 24 hpi), which results in pre-haustorial resistance in the CIFC 832/1 genotype [27].

In addition to increasing our knowledge about this pathosystem, the cytological analysis allowed the selection of the most suitable key time points of the infection process that can better understand transcriptome analysis. To study differentially expressed genes in the interaction between coffee and the race XXXIII of *H. vastatrix*, we considered the defined time points, 12, 24, and 96 hpi, and 17 dpi for the construction of cDNA libraries. Based on the cDNA libraries, we obtained a transcriptome with about 162 million trimmed reads. The number and average length of these transcripts were higher than those previously obtained [28]. In the previous study, a large-scale transcriptome analysis of coffee–*H. vastatrix* interaction was conducted at a late time point (21 dpi), and 22,700 contigs with an average length of 426 bp were assembled. Most of them were from coffee and only 6763 contigs were assigned to *H. vastatrix*. In addition, plant and fungi reads were not separated before assembly, and chimera contigs may have been generated. Here, we analyzed early events of the infection process in both interactions, and separated the transcripts from both organisms, which allows a more comprehensive view of the fungal transcriptome resulting in a higher number of transcripts from the pathogen. The transcriptome obtained from the race XXXIII of *H. vastatrix* consisted of 29,812 transcripts, with an average length of 1463 nucleotides. Mapping analysis of the trimmed reads in the coffee and *H. vastatrix* reference genomes showed that the amount of reads from the fungus in each library is correlated with infection progression. Since an intensive growth of intercellular hyphae and haustoria was observed at 17 dpi, this result reflects the establishment of infection in the susceptible genotype. Thus, an increased fungal biomass led to a high amount of fungal mRNA, as expected. Similar results have been found for other rust-causing fungi such as *Puccinia striiformis* f. sp. *tritici*, which causes wheat yellow rust [56].

After separating the transcripts from *H. vastatrix*, the transcriptome obtained from the interaction libraries was compared with the transcriptome of the uredospore libraries (HU and GU). The comparative study indicated that, for obligate parasites such as *H. vastatrix*, in vitro studies (e.g., germinated uredospores) do not reflect gene expression changes in vivo. The use of uredospores as the source of material for RNA sequencing, as reported for *P. helianthi* [57], is an important initial step in a study; however, it may not comprise all the biological information, especially when searching for new effectors. Moreover, several differentially expressed genes were found at early time points of both interactions; however, the major driving force behind the changes in global gene expression may be related to different stages in the fungus life cycle (e.g., haustorial differentiation, sporulation, etc.) instead of different fungus "strategies" to fight back plant defense.

To identify DEGs, all 10 cDNA libraries were compared. The results suggested that, when the fungus interacts with a susceptible plant, a greater set of genes is upregulated in comparison to a resistant plant. To establish a biotrophic interaction, the fungus needs to suppress or evade host defense reactions [16]. Different genes play a key role in this process, preventing an efficient counterattack of the host. The high number of *H. vastatrix* upregulated genes at early stages of the infection process constitutes a strategy of the pathogen to protect itself, avoiding recognition by the plant and hyphal lysis by extracellular plant chitinases, which enables the colonization of the host [16,58]. However, the fungus ceases to grow in the early stages of infection when interacting with a resistant plant, with the disruption of the cytoplasmic contents of fungal infection structures [36], as shown in our cytological analysis. Furthermore, the results suggested that the fungal transcriptome is dynamically altered over the course of the infection.

The identified DEGs were functionally annotated; the low number of matches with transcripts of *H. vastatrix* reflects the high diversity among *H. vastatrix* races and highlights the lack of information on this species. The high number of *H. vastatrix* transcripts matching *M. larici-populina* and *P. graminis* has also been reported previously [28].

In the annotation, the most representative category was "signaling," followed by "protein metabolism" and "transport." Studies of the *H. vastatrix* transcriptome, mainly at later stages of infection, have shown the expression of genes involved mainly in signaling and the establishment and maintenance of biotrophy. These genes include some candidate effectors such as homologues of haustorially expressed secreted proteins (HESPs) and rust transferred protein 1 (RTP1), ketin deacetylases, and endoglucanases [28,29,59]. In germinating spores, intense transport, secretory activity, and cellular multiplication have been verified, while in appressoria the expressed genes are related to active metabolism, translational activity, production of new structures, and signaling [29]. These observations suggest that the communication between the plant and fungus starts early, even prior to penetration.

The findings of our study and those of others on *H. vastatrix* suggest that the fungus constantly modulates its transcriptome, transducing signals to complete its life cycle. This wide range of categories with DEGs reinforces the notion that several cellular functions may be important for an efficient infection. In fact, the suppression of non-related genes such as amino acid permease (transport), secreted glycosyl hydrolase, a predicted glycolytic enzyme (carbohydrate metabolism), and a gene of unknown function has been reported to impair rust development by *Puccinia* species [60].

By comparing the libraries of each time point from both interactions, we found that the putative secreted effectors do not differ significantly between interactions. The results indicated that the infection strategy is similar in both cases, regardless of the plant response. This finding was also verified with the PCoA and hierarchical clustering analyses. On the other hand, a previous study [27] conducting the same type of analysis for the coffee transcriptome, not the fungus transcriptome, showed more changes in the transcriptome of the resistant coffee genotype (CIFC 832/1), especially in the early events, than in that of the susceptible one (Caturra CIFC 19/1).

Furthermore, the transcriptome of *H. vastatrix* that we obtained was also used to mine for putative pathogen effectors using a co-expression network analysis. This strategy relies on the guilt-by-association principle, in which genes with similar expression patterns tend to be related to the same biological process [61]. In this analysis, the effector HvEC-016 from *H. vastatrix* race II, identified previously as the avirulence gene corresponding to the resistant gene $S_H1$ [44], was included as a bait. We found one transcript, 2156, homologue to this *H. vastatrix* gene. Moreover, some mutations were identified between the sequences of the homologue genes. This is an important finding because the genes from race XXXIII ($v_{5,7}$ or $v_{5,7,9}$) are not recognized by the $S_H1$ gene [1,12], which suggests that these few mutations can be responsible for the avoidance of plant resistance through the $S_H1$ gene. The found variations could be used to develop molecular markers that can help differentiate races. This has been done only based on plant inoculation methods, which are time-consuming and may achieve inconclusive results [1].

Based on all analyses of the *H. vastatrix* transcriptome, we selected 12 genes (11 DEGs and transcript 2156) that might be potential effectors and evaluated them using qPCR. Eight genes were upregulated in the early stages and were considered the best effector candidates. The upregulated expression of putative secreted effectors at early events of the infection has been reported to prevent the induction of the HR by *Phakopsora pachyrhizi* in susceptible plants [62], highlighting the function of those genes as effectors. Our qPCR results also suggested that the putative effectors are expressed independently of the plant responses, which is similar to the results obtained by PCoA and DEG analyses. Effectors are key determinants of pathogenicity, and the pathogen must coordinate its arsenal by up- or downregulating its genes to colonize the host tissues. Genes involved in the detoxification of defense compounds [54], suppression of the host immune system [63,64], and assimilation of nutrients during colonization [65,66] must be upregulated, whereas other genes must be downregulated to avoid recognition by the host. The identification of signals or signaling pathways that regulate effector gene expression offers another dimension for the development of new management strategies.

## 6. Conclusions

Despite the difficulty of studying a non-model obligatory parasite with a highly complex genome, we were able to produce a good quality transcriptome of *H. vastatrix* (race XXXIII). Fungal genes that are expressed at key time points of the infection in coffee, considering compatible and incompatible interactions, are now available. The data allows to identify fungal DEGs in both interactions, and fungal candidate effectors were identified. The data of gene expression during the infection process contribute to elucidating the molecular mechanisms leading to the breakdown of coffee resistance by new physiological races of the fungus. Thus, this study provides new insights into the coffee–*H. vastatrix* interaction involving the race XXXIII of fungus and represents an important step towards understanding the infection process of rust disease.

**Supplementary Materials:** The following supporting information can be downloaded at: www.mdpi.com/article/10.3390/agronomy12020444/s1, Figure S1: Bioinformatic pipeline analysis reads processing, mapping to reference genomes, filtering plant reads from fungi reads, *H. vastatrix* transcriptome assembly and identification of differentially expressed genes in all sequenced libraries; Figure S2: Blastx search against NCBI-NR (A) and PHI (B) databases. All *H. vastatrix* DEGs were analyzed by blastx and the species of first hit of each DEG was retrieved; Table S1: Functional annotation of the identified DEGs in the *Hemileia vastatrix* transcriptome; Table S2: Transcripts identified in a co-expression network with transcript 2156.

**Author Contributions:** Conceptualization E.T.C., E.M.Z. and L.Z.; methodology, R.d.L.F.-L., S.d.S.F., I.S.L.C., and J.C.F.; data analysis, R.d.L.F.-L., S.d.S.F. and T.E.F.M.; validation and formal analysis, E.T.C. and E.M.Z.; investigation, I.S.L.C., R.d.L.F.-L., S.S.F. and E.T.C.; writing—original draft preparation, R.d.L.F.-L., I.S.L.C., S.d.S.F., and J.C.F.; writing—review and editing, I.S.L.C., J.C.F. and E.T.C.; supervision, E.T.C. and E.M.Z.; funding acquisition, E.T.C., E.M.Z. and L.Z. All authors have read and agreed to the published version of the manuscript.

**Funding:** This work was financially supported by the Brazilian Coffee Research and Development Consortium (Consórcio Pesquisa Café, CBP&D/Café), the Foundation for Research Support of the State of Minas Gerais (FAPEMIG), the National Council of Scientific and Technological Development (CNPq), the National Institutes of Science and Technology of Coffee (INCT/Café), and Coordination for the Improvement of Higher Education Personnel (CAPES).

**Institutional Review Board Statement:** Not applicable.

**Informed Consent Statement:** Not applicable.

**Data Availability Statement:** Transcriptome datasets are available in the National Center for Biotechnology Information under BioProject PRJNA796568, PRJNA353185, and PRJNA353233.

**Conflicts of Interest:** The authors have no conflict of interest to declare and confirm that each one has made substantial contributions to the information or materials submitted for publication.

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
