# Peer review of "Transcriptome Analysis Uncovers the Gene Expression Profile of Hemileia vastatrix (Race XXXIII) during the Interactions with Resistant and Susceptible Coffee"

_agronomy, doi:10.3390/agronomy12020444_

Round 1
Reviewer 1 Report
The work performed is useful to the plant-pathogen interaction studies. The identification of differentially expressed genes offer a huge potential for researches related to interaction between the fungus and coffee. However, However, the conclusions do not seem to give the proper focus of the work carried out. The approach seems more discussion of results. It is suggested to improve this section. Please also check the English and typos. Make these changes so that the work can be published.
Author Response
"Please see the attachment."

Reviewer 2 Report
Coffee is an important economic crop and leaf rust is a destructive disease affecting safe production of the crop. Analysis of hostresistnace against the rust pathogen based on molecular level can be provided an insight into understanding the mechanism of the host to the leaf rust pathogen and improving host resistance. There are some comments and suggestions that made in revised PDF texts for this manuscript.

Author Response
"Please see the attachment."
